# End-to-End Training of Multi-Document Reader and Retriever for Open-Domain Question Answering

**Devendra Singh Sachan**[1,2]**, Siva Reddy**[1,2]**, William Hamilton**[1,2]**, Chris Dyer**[3]**, Dani Yogatama**[3]

[1]Mila - Quebec AI Institute
[2]School of Computer Science, McGill University
[3]DeepMind
sachande@mila.quebec, {siva, wlh}@cs.mcgill.ca
{cdyer, dyogatama}@deepmind.com

## Abstract

We present an end-to-end differentiable training method for retrieval-augmented open-domain question answering systems that combine information from multiple retrieved documents when generating answers. We model retrieval decisions as latent variables over sets of relevant documents. Since marginalizing over sets of retrieved documents is computationally hard, we approximate this using an expectation-maximization algorithm. We iteratively estimate the value of our latent variable (the set of relevant documents for a given question) and then use this estimate to update the retriever and reader parameters. We hypothesize that such end-to-end training allows training signals to flow to the reader and then to the retriever better than stage-wise training. This results in a retriever that is able to select more relevant documents for a question and a reader that is trained on more accurate documents to generate an answer. Experiments on three benchmark datasets demonstrate that our proposed method outperforms all existing approaches of comparable size by 2-3 absolute exact match points, achieving new state-of-the-art results. Our results also demonstrate the feasibility of learning to retrieve to improve answer generation without explicit supervision of retrieval decisions.

## 1 Introduction

Open-domain question answering (OpenQA) is a question answering task where the goal is to train a language model to produce an answer for a given question. In contrast to many question answering tasks, an OpenQA model is only provided with the question as its input without accompanying documents that contain the answer. One of the most promising approaches to OpenQA is based on augmenting the language model with an external knowledge source such as Wikipedia (often referred to as the evidence documents). In this approach, the model consists of two core components (Chen et al., 2017): (i) an information retrieval system to identify useful pieces of text from the knowledge source (the retriever); and (ii) a system to produce the answer given the retrieved documents and the question (the reader).

We can view such a model as a latent variable model, where the latent variables represent retrieved documents that are used to produce answers given questions (Lee et al., 2019). End-to-end (joint) training of this model is challenging since we need to learn both to generate an answer given retrieved documents and what to retrieve. Previous work considers two potential solutions (see Table 1 for a high-level summary). First, they adopt a stage-wise training, where the retriever is trained while freezing the reader and vice versa (Karpukhin et al., 2020, Izacard and Grave, 2021b,a). Another

35th Conference on Neural Information Processing Systems (NeurIPS 2021).

| | Reader and Retriever Training | | | | | |
|---|---|---|---|---|---|---|
| **Model** | *Multi-Doc Reader* | *Retriever Adaptation* | *Disjoint* | *End-to-End* | *Multi-Step* | *Unsupervised Retriever* |
| REALM (Guu *et al.*, 2020) | | ✓ | | ✓ | | ✓ |
| DPR (Karpukhin *et al.*, 2020) | | | ✓ | | | |
| RAG (Lewis *et al.*, 2020b) | | ✓ | | ✓ | | |
| FiD (Izacard and Grave, 2021b) | ✓ | | ✓ | | | |
| FiD-KD (Izacard and Grave, 2021a) | ✓ | ✓ | | | ✓ | |
| EMDR$^2$ (Our Approach) | ✓ | ✓ | | ✓ | | ✓ |

**Table 1:** Bird's-eye view of the recent OpenQA approaches. **Multi-Doc reader** indicates whether the reader architecture uses multiple documents or a single document. **Retriever adaptation** shows whether the retriever gets feedback from the reader to update its parameters. **Disjoint** denotes that first the retriever is trained and then the reader is trained. **End-to-end** denotes that the reader and retriever are trained jointly in one cycle. **Multi-step** indicates that the reader and retriever are trained iteratively in multiple cycles. **Unsupervised retriever** indicates whether the retriever is initialized using unsupervised approaches or using supervised data.

alternative is to constraint the reader to condition on each retrieved document individually[1] (Guu *et al.*, 2020)—sometimes with extra supervision for the latent variables in the form of the relevant document for a question (Lewis *et al.*, 2020b).

In this paper, we consider a retrieval-augmented question answering model that combines information from multiple documents when generating answers. Expectation-maximization (Dempster *et al.*, 1977) offers a principled template for learning this class of latent variable models. We present EMDR$^2$: **E**nd-to-end training of **M**ulti-**D**ocument **R**eader and **R**etriever (§2). EMDR$^2$ iteratively uses feedback from the model itself as "pseudo labels" of the latent variables for optimizing the retriever and reader parameters. We use two estimates of the latent variables: (i) prior scores for updating the reader parameters and (ii) approximate posterior scores given all observed variables for the retriever parameters.

We evaluate our proposed method by experimenting on three commonly used OpenQA datasets: Natural Questions, TriviaQA, and WebQuestions (§3). EMDR$^2$ achieves new state-of-the-art results for models of comparable size on all datasets, outperforming recent approaches by 2-3 absolute exact match points. We also show that EMDR$^2$ is robust to retriever initialization. It achieves high accuracy with unsupervised initialization, suggesting that supervised training of the retriever may not be an essential component of the training process as suggested in prior work (Karpukhin *et al.*, 2020).

In summary, our contributions are as follows: (i) we present an end-to-end training method (EMDR$^2$) for retrieval-augmented question-answering systems; (ii) we demonstrate that EMDR$^2$ outperforms other existing approaches of comparable size without any kind of supervision on the latent variables; (iii) we provide ablation studies for a better understanding of the contributions of different components of our proposed method; and (iv) we release our code and checkpoints to facilitate future work and for reproducibility.[2]

EMDR$^2$ is a framework that can be used to train retrieval-augmented text generation models for any task. We believe that our estimation technique in EMDR$^2$ is also useful for learning similar latent variable models in other domains.

## 2 Model

Our proposed model EMDR$^2$ consists of two components: (i) a neural retriever and (ii) a neural reader, which we train jointly in an end-to-end setting. Figure 1 shows an illustration of our model and training procedure. We discuss each component and our training objective in detail below.

---

[1]This makes marginalization over the latent variables easier since we only need to consider one document at a time rather than multiple documents at once.

[2]Our code is available at: `https://github.com/DevSinghSachan/emdr2`

## 2.1 Neural Retriever: Dual Encoder

Let the collection of evidence documents be denoted by $\mathcal{D} = \{\boldsymbol{d}_1, \ldots, \boldsymbol{d}_M\}$. Given a question $\boldsymbol{q}$, the goal of the retriever module is to select a subset of documents $\mathcal{Z} \subset \mathcal{D}$ to answer the question. We model the retriever as a dual-encoder network (Bromley *et al.*, 1994), where one encoder $f_q$ encodes the question and another $f_d$ encodes the evidence document (to a vector). The retrieval score is defined as the dot product between the two resulting vectors:

$$\text{score}(\boldsymbol{q}, \boldsymbol{d}_i; \Phi) = f_q(\boldsymbol{q}; \Phi_q)^\top f_d(\boldsymbol{d}_i; \Phi_d), \tag{1}$$

where $\Phi = [\Phi_q, \Phi_d]$ denotes the retriever parameters. We select top-$K$ documents for the question $\boldsymbol{q}$ from $\mathcal{D}$ based on the retrieval scores. We denote the set of retrieved documents by $\mathcal{Z} = \{\boldsymbol{z}_1, \ldots, \boldsymbol{z}_K\}$.

We use transformer encoders (Vaswani *et al.*, 2017) as our $f_q$ and $f_d$. Our transformer architecture is similar to BERT with 12 layers and 768 hidden size (Devlin *et al.*, 2019). We use the final representation of the first token (i.e., the standard [CLS] token from BERT's tokenization) as our question (and similarly document) embedding. Initializing $f_q$ and $f_d$ with BERT weights has been shown to lead to a poor retrieval accuracy (Lee *et al.*, 2019, Sachan *et al.*, 2021). Therefore, we initialize the retriever with an unsupervised training procedure. We discuss our initialization technique in detail in §3.2.

## 2.2 Neural Reader: Fusion-in-Decoder

The reader takes as input a question $\boldsymbol{q}$ and a set of retrieved documents (to be read) $\mathcal{Z}$ to generate an answer. Our reader is based on the Fusion-in-Decoder (FiD; Izacard and Grave, 2021b) model, which is built on top of T5 (Raffel *et al.*, 2020). T5 is a pretrained sequence-to-sequence transformer that consists of an encoder $g_e$ and a decoder $g_d$.

In FiD, each retrieved document $\boldsymbol{z}_k$ is first appended with its title ($\boldsymbol{t}_{\boldsymbol{z}_k}$) and the question:

$$\boldsymbol{x}_k = \text{[CLS]}\,\boldsymbol{q}\,\text{[SEP]}\,\boldsymbol{t}_{\boldsymbol{z}_k}\,\text{[SEP]}\,\boldsymbol{z}_k\,\text{[SEP]},$$

where [CLS] is used to indicate the start of a document and [SEP] is used as a separator for the different parts of the document as well as the final token.

Each $\boldsymbol{x}_k$ is then independently given as an input to the T5 encoder $g_e$. The output representations corresponding to all of the retrieved documents are concatenated as:

$$\mathbf{X}_{\mathcal{Z}} = [g_e(\boldsymbol{x}_1); \ldots; g_e(\boldsymbol{x}_K)] \in \mathbb{R}^{(N \times K) \times H},$$

where $N$ is the number of tokens in each $\boldsymbol{x}_k$[3] and $H$ is the hidden size of the T5 encoder $g_e$. In this work, we use the T5-*base* configuration with $N = 512$ and $H = 768$.

$\mathbf{X}_{\mathcal{Z}}$ is then given as an input to the T5 decoder $g_d$. When generating an answer token, the decoder attends to both previously generated tokens (i.e., causal attention) as well as the tokens encoded in $\mathbf{X}_{\mathcal{Z}}$ (i.e., cross attention). Since $\mathbf{X}_{\mathcal{Z}}$ contains information from multiple documents, the decoder has the ability to aggregate useful signals contained in multiple documents and jointly reason over them. We define the probability of the answer as:

$$p(\boldsymbol{a} \mid \boldsymbol{q}, \mathcal{Z}; \Theta) = \prod_{t=1}^{T} p\left(a_t \mid \boldsymbol{a}_{<t}, \boldsymbol{q}, \mathcal{Z}; \Theta\right), \tag{2}$$

where $\Theta$ denotes the reader parameters (i.e., T5 encoder and decoder) and $T$ is the number of answer tokens. We keep generating answer tokens until the decoder outputs a special EOS token or a pre-specified maximum answer length is reached.

## 2.3 End-to-End Training of Reader and Retriever

In contrast to previous work on generative question answering, we train both the reader and the retriever jointly in an end-to-end differentiable fashion.

Denote our latent variable which represents a set of retrieved documents by $Z$ and let $\mathcal{Z}$ be a possible value of $Z$. The marginal likelihood of an answer (marginalizing over all the possible values of $Z$)

---

[3]We truncate and pad as necessary such that every $\boldsymbol{x}_k$ has the same length $N$. See §3.2 for details.

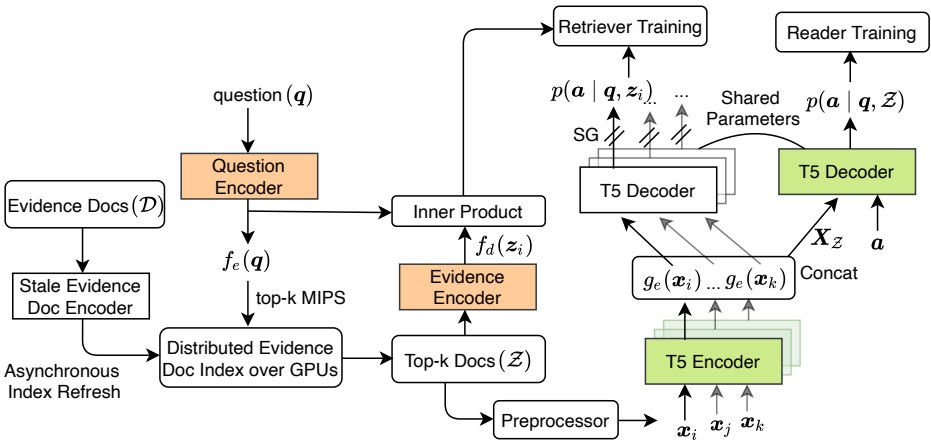

**Figure 1:** An illustration of the different components of $\text{EMDR}^2$. Colored blocks indicate components which contain trainable parameters.

is: $p(\boldsymbol{a} \mid \boldsymbol{q}; \Theta, \Phi) = \sum_{Z=\mathcal{Z}} p(\boldsymbol{a} \mid \boldsymbol{q}, \mathcal{Z}; \Theta) p(\mathcal{Z} \mid \boldsymbol{q}; \Phi)$. The goal of our training procedure is to find $\Phi$ and $\Theta$ that would maximize the above objective. Exactly optimizing Eq. 3 is intractable as it is combinatorial in nature.[4] For one particular value $\mathcal{Z}$, the log-likelihood is simpler to compute: $\log p(\boldsymbol{a} \mid \boldsymbol{q}, \mathcal{Z}; \Theta) p(\mathcal{Z} \mid \boldsymbol{q}; \Phi) = \log p(\boldsymbol{a} \mid \boldsymbol{q}, \mathcal{Z}; \Theta) + \log p(\mathcal{Z} \mid \boldsymbol{q}; \Phi)$.

Expectation-maximization (EM) algorithm (Dempster *et al.*, 1977) offers a solution to learning this latent variable model. In classical EM, we iteratively compute the posterior of $Z$ given all observed variables and use it to update $\Theta$ and $\Phi$.

We propose using two estimates of $Z$—$\mathcal{Z}_{\text{reader}}$ and $\mathcal{Z}_{\text{retriever}}$—for updating the two components of the model (reader parameters $\Theta$ and retriever parameters $\Phi$):

$$\log \underbrace{p(\boldsymbol{a} \mid \boldsymbol{q}, \mathcal{Z}_{\text{reader}}; \Theta)}_{\text{reader}} + \log \underbrace{p(\mathcal{Z}_{\text{retriever}} \mid \boldsymbol{q}; \Phi)}_{\text{retriever}}. \tag{3}$$

In the first term, we set the value of the latent variable $Z = \mathcal{Z}_{\text{reader}}$ based on the prior scores. In the second term, we seek to maximize an approximate posterior of $Z = \mathcal{Z}_{\text{retriever}}$. We discuss them in more detail below.

**Reader parameters $\Theta$.** For updating $\Theta$ (the first term of Eq. 3), we use the top-$K$ documents with the highest individual scores (as computed by Eq. 1 based on the current value of $\Phi$) to construct $\mathcal{Z}_{\text{reader}}$. This is equivalent to relying on the prior $p(Z \mid \boldsymbol{q}; \Phi)$ to estimate $\mathcal{Z}_{\text{reader}}$ (without using information from the answer $\boldsymbol{a}$). We choose to use the prior to train reader parameters since the prior scores are also used at evaluation time to obtain the top-$K$ documents. As a result, there is no mismatch between training and test computations when computing $p(\boldsymbol{a} \mid \boldsymbol{q}, \mathcal{Z}; \Theta)$ (i.e., $\mathcal{Z}$ that is used at test time is obtained in exactly the same way as $\mathcal{Z}_{\text{reader}} = \mathcal{Z}_{\text{top-}K}$).

**Retriever parameters $\Phi$.** For updating $\Phi$ (the second term of Eq. 3), we propose to use the posterior estimate. In other words, we use additional information from $\boldsymbol{a}$ when evaluating $Z_{\text{retriever}}$ to train $\Phi$. Using the posterior allows our retriever to learn from richer training signals as opposed to relying only on the prior.

We need to be able to compute $p(\mathcal{Z}_{\text{retriever}} \mid \boldsymbol{q}, \boldsymbol{a}; \Theta, \Phi)$ to maximize the retriever parameters. However, computing this quantity is difficult since it is a probability of a set.[5] Consider a set of $K$ documents (e.g., $\mathcal{Z}_{\text{top-}K}$), where $\boldsymbol{z}_k$ denotes a document in the set. We approximate the maximization of the probability of the set by assuming that its probability is maximized if the sum of the probability of

---

[4]Contrast our objective with REALM (Guu *et al.*, 2020), where the reader only conditions on one retrieved document $\boldsymbol{z}_k$ when generating an answer. In this case, the latent variable represents a document assignment instead of a set of retrieved documents.

[5]This is true whether we choose to use the posterior probability or the prior probability.

each document in the set is maximized.[6] With this approximation, we arrive at a simpler quantity: $\sum_{k=1}^{K} p(\boldsymbol{z}_k \mid \boldsymbol{q}, \boldsymbol{a}; \Theta, \Phi)$. Note that using Bayes rule, we can rewrite:[7]

$$p(\boldsymbol{z}_k \mid \boldsymbol{q}, \boldsymbol{a}; \Theta, \Phi) \propto p(\boldsymbol{a} \mid \boldsymbol{q}, \boldsymbol{z}_k; \Theta) p(\boldsymbol{z}_k \mid \boldsymbol{q}; \Phi). \tag{4}$$

The reader now only conditions on one document when computing the probability of an answer $p(\boldsymbol{a} \mid \boldsymbol{q}, \boldsymbol{z}_k; \Theta)$. This simpler reader uses the same parameters as the more sophisticated one $\Theta$, but it only uses one document $\boldsymbol{z}_k$ instead of a set of documents.

To compute Eq. 4, we first obtain $K$ documents with the highest scores as computed by Eq. 1 based on the current value of $\Phi$. We compute the probability of document $\boldsymbol{z}_k \in \mathcal{Z}_{\text{top-}K}$ as:

$$p(\boldsymbol{z}_k \mid \boldsymbol{q}, \mathcal{Z}_{\text{top-}K}; \Phi) \approx \frac{\exp(\text{score}(\boldsymbol{q}, \boldsymbol{z}_k)/\tau; \Phi)}{\sum_{j=1}^{K} \exp(\text{score}(\boldsymbol{q}, \boldsymbol{z}_j)/\tau; \Phi)}, \tag{5}$$

where $\tau$ is a temperature hyperparameter and the approximation assumes that documents beyond the top-$K$ contributes very small scores so we do not need to sum over all evidence documents $M$ in the denominator (which is in the order of tens of millions in our experiments). We then compute $p(\boldsymbol{a} \mid \boldsymbol{q}, \boldsymbol{z}_k; \Theta)$ similarly to Eq. 2.

**Overall training objective of EMDR$^2$.** Combining the above derivations, our end-to-end training objective that we seek to maximize for a particular example becomes:

$$\mathcal{L} = \underbrace{\log p(\boldsymbol{a} \mid \boldsymbol{q}, \mathcal{Z}_{\text{top-}K}; \Theta)}_{\text{reader}} + \underbrace{\log \sum_{k=1}^{K} \mathbb{SG}\left(p(\boldsymbol{a} \mid \boldsymbol{q}, \boldsymbol{z}_k; \Theta)\right) p(\boldsymbol{z}_k \mid \boldsymbol{q}, \mathcal{Z}_{\text{top-}K}; \Phi)}_{\text{retriever}}, \tag{6}$$

where $\mathbb{SG}$ is the stop-gradient operator so that the reader parameters $\Theta$ are not updated to also perform well given a single document $\boldsymbol{z}_k$. The stop-gradient operator in the second term of EMDR$^2$ has several benefits. First, the FiD reader is trained from the first term of the EMDR$^2$ objective in which its likelihood is conditioned on all the retrieved documents, similar to how the reader is used at test time. Second, it also makes training faster since the backward pass which is computationally more expensive than the forward pass is not needed, which in turn reduces the usage of GPU RAM as intermediate activations need not be saved.

Given a training example, we update $\Theta$ and $\Phi$ by taking gradients of Eq. 6 with respect to $\Theta$ and $\Phi$ in an end-to-end fashion. Intuitively, we train the reader to generate the correct answer given $K$ highest scoring documents $\mathcal{Z}_{\text{top-}K}$. For the retriever, we train it to select $K$ documents which *collectively* has a high score of generating an answer (since the sum over $K$ is inside the log in the second term) while taking into account feedback from the reader. Algorithm 1 summarizes our training algorithm.

---

**Algorithm 1:** End-to-end training of multi-document reader and retriever.

---

**Input:** Model parameters $\Theta$ and $\Phi$, evidence documents $\mathcal{D}$.
**while** *not converged* **do**
  - Compute $\mathcal{Z}_{\text{top-}K}$ using the current retriever parameters $\Phi$.      // E-step
  - Compute $p(\boldsymbol{a} \mid \boldsymbol{q}, \boldsymbol{z}_k)$ for each $\boldsymbol{z}_k$ using the current reader parameters $\Theta$.    // E-step
  - Update model parameters $\Theta$ and $\Phi$ to maximize the log-likelihood in Eq. 6. // M-step
**end**

---

## 3 Experiments

### 3.1 Datasets

We experiment with three commonly used open-domain question answering datasets:

---

[6]The intuition is that each element of the set contributes independently, which greatly simplifies the computation to find the maximum of the set.

[7]We choose not to normalize with $p(\boldsymbol{a} \mid \boldsymbol{q}; \Theta, \Phi)$ since computing this quantity would require summing over all evidence documents $M$. While this makes the resulting objective that we optimize not correspond to a proper probability distribution anymore, we observe that our training method still behaves well in practice.

- **Natural Questions (NQ; Kwiatkowski *et al.*, 2019).** NQ contains questions asked by users of the Google search engine. Similar to Lee *et al.* (2019), we use the short answer subset.
- **TriviaQA (Joshi *et al.*, 2017).** TriviaQA is a collection of trivia question-answer pairs that were collected from multiple sources on the web.
- **WebQuestions (WebQ; Berant *et al.*, 2013).** WebQ questions were collected using Google Suggest API and the answers were annotated using Mechanical Turk. We use the version from Chen *et al.* (2017) where Freebase IDs in the answers are replaced by entity names.

**Evidence documents $\mathcal{D}$.** We use the preprocessed English Wikipedia dump from December 2018 released by Karpukhin *et al.* (2020) as our evidence documents. Each Wikipedia article is split into non-overlapping 100 words long segments. Each segment corresponds to a document in our case. There are a total of 21,015,324 documents in total.

We provide descriptive statistics and other preprocessing details in Appendix A.

### 3.2 Implementation Details

**Hardware and library.** We run all of our experiments on a machine with 96 CPUs, 1.3TB physical memory, and 16 A100 GPUs. We use PyTorch (Paszke *et al.*, 2019) to implement our proposed model and relevant baselines.

**Model configurations.** For both the retriever and reader, we use the *base* configuration that consists of 12 layers, 768 dimensional hidden size, and 12 attention heads. In all experiments, we retrieve 50 documents, unless stated otherwise. We only use the base configuration in our experiments due to GPU memory constraints. However, we believe that our results would generalize to larger configurations as well.

**Retrieval.** To support fast retrieval, we pre-compute evidence document embeddings and store them in a distributed fashion over all the GPUs. We refer to these document embeddings as the document index. For each question, we retrieve documents in an online (on-the-fly) manner by performing exact maximum inner product search (MIPS), implemented using asynchronous distributed matrix multiplication over the document index. These documents are converted to subwords using BERT's tokenization and are given as input to the T5 reader. If a tokenized document is shorter than 512 tokens, it is padded using the tokens from the neighboring documents until the maximum token limit is reached. Such padding additionally helps to provide an extended context for answer generation.

**Initialization and training details.** We initialize the parameters of the model with unsupervised pre-training before performing supervised training using the question-answer training examples. Unsupervised pre-training is essential as it helps to warm-start the retriever so that it outputs relevant documents for a given question.

We first pre-train the retriever parameters with unsupervised Inverse Cloze Task training (Lee *et al.*, 2019) for 100,000 steps. We then extract sentences containing named entities from the evidence documents. Next, we replace 15% of the named entity tokens with masked tokens, which are often referred to as masked salient spans (MSS; Guu *et al.*, 2020). The masked sentence can be considered as the question and its salient spans (i.e, named entities) can be considered as the answer to train the model with Eq. 6. We train the model on these question-answer (masked sentence-named entities) pairs for 82,000 steps with a batch size of 64 using Adam (Kingma and Ba, 2015). We refer to this initialization method as *unsupervised pre-training with masked salient spans*. We provide further description in Appendix C.

After MSS training, we finetune the model on the dataset-specific question-answer training examples with EMDR$^2$. We perform training for 10 epochs on NQ and TriviaQA with a batch size of 64, and for 20 epochs on WebQ with a batch size of 16. During training, we save a checkpoint every 500 steps and select the best checkpoint based on its performance on the development set.

During end-to-end training, since the parameters of the document encoder ($f_d$) are also updated at every step, the pre-computed document embeddings become stale as training progresses. We use the most recent document encoder checkpoint to compute fresh document embeddings asynchronously with which the document index is updated after every 500 training steps to prevent staleness.

| Model | top-$K$ | NQ dev | NQ test | TriviaQA dev | TriviaQA test | WebQ dev | WebQ test | *# of params* |
|---|---|---|---|---|---|---|---|---|
| **Closed-Book QA Models** | | | | | | | | |
| T5-*base* (Roberts *et al.*, 2020) | 0 | - | 25.7 | - | 24.2 | - | 28.2 | 220M |
| T5-*large* (Roberts *et al.*, 2020) | 0 | - | 27.3 | - | 28.5 | - | 29.5 | 770M |
| T5-*XXL* (Roberts *et al.*, 2020) | 0 | - | 32.8 | - | 42.9 | - | 35.6 | 11B |
| GPT-3 (Brown *et al.*, 2020) | 0 | - | 29.9 | - | - | - | 41.5 | 175B |
| **Open-Book QA Models** | | | | | | | | |
| BM25 + BERT (Lee *et al.*, 2019) | 5 | 24.8 | 26.5 | 47.2 | 47.1 | 27.1 | 21.3 | 220M |
| ORQA (Lee *et al.*, 2019) | 5 | 31.3 | 33.3 | 45.1 | 45.0 | 36.8 | 30.1 | 330M |
| REALM (Guu *et al.*, 2020) | 5 | 38.2 | 40.4 | - | - | - | 40.7 | 330M |
| DPR (Karpukhin *et al.*, 2020) | 25 | - | 41.5 | - | 56.8 | - | 34.6 | 330M |
| RECONSIDER (Iyer *et al.*, 2021)† | 30 | - | 43.1 | - | 59.3 | - | 44.4 | 440M |
| RAG-Sequence (Lewis *et al.*, 2020b)† | 50 | 44.0 | 44.5 | 55.8 | 56.8 | 44.9 | 45.2 | 626M |
| Individual Top-$K$ (Sachan *et al.*, 2021) | - | - | 45.9 | - | 56.3 | - | - | 440M |
| Joint Top-$K$ (Sachan *et al.*, 2021) | 50 | - | 49.2 | - | 64.8 | - | - | 440M |
| FiD (Izacard and Grave, 2021b) | 100 | - | 48.2 | - | 65.0 | - | - | 440M |
| FiD-KD (Izacard and Grave, 2021a) | 100 | 48.0 | 49.6 | 68.6 | 68.8 | - | - | 440M |
| **Our Implementation (Base Configuration)** | | | | | | | | |
| FiD / T5-*base* | 0 | 26.0 | 25.1 | 26.7 | 27.8 | 31.0 | 32.4 | 220M |
| FiD (DPR retriever, T5 reader) | 1 | 37.3 | 38.4 | 50.8 | 50.4 | 40.2 | 38.3 | 440M |
| FiD (DPR retriever, T5 reader) | 50 | 47.3 | 48.3 | 65.5 | 66.3 | 46.0 | 45.2 | 440M |
| FiD (MSS + DPR retriever, T5 reader) | 50 | 48.8 | 50.4 | 68.0 | 68.8 | 43.5 | 46.8 | 440M |
| FiD (MSS retriever, MSS reader) | 50 | 38.5 | 40.1 | 60.0 | 59.8 | 39.1 | 40.2 | 440M |
| EMDR$^2$ (MSS retriever, MSS reader) | 50 | **50.4** | **52.5** | **71.1** | **71.4** | **49.9** | **48.7** | 440M |

**Table 2:** Exact match scores on three evaluation datasets. Top-$K$ denotes the number of retrieved documents that are used by the reader to produce an answer. To provide a fair comparison with our reimplementations, we show results from other papers with the base configuration, except for RAG-Sequence that uses BART-*large* (Lewis *et al.*, 2020a). † indicates that their results on WebQ use NQ training data to pretrain the model.

**Inference.** We use greedy decoding for answer generation at inference time.

## 3.3 Baselines

We compare our model to other approaches for OpenQA that can be categorized under the following two classes:

- **Closed-book QA models.** Large-scale language models capture a lot of world knowledge in their parameters derived from the corpus they have been trained on (Petroni *et al.*, 2019). We compare with the work of Roberts *et al.* (2020) who show that larger T5 models—when finetuned with question-answer pairs—can perform remarkably well. We also compare with the few-shot results of GPT-3 (Brown *et al.*, 2020).[8]

- **Open-book QA models.** Similar to this work, these models consist of retriever and reader components and adopt the retrieve then predict approach for answering questions given a collection of evidence documents. These models mainly differ in how the retriever is initialized (ORQA; Lee *et al.*, 2019, DPR; Karpukhin *et al.*, 2020), whether the reader processes a single document (ORQA, DPR, RAG; Lewis *et al.*, 2020b) or multiple documents (FiD; Izacard and Grave, 2021b), or whether the reader and retriever are trained jointly or in a multistage process (REALM; Guu *et al.*, 2020, FiD-KD; Izacard and Grave, 2021a).

## 3.4 Results

We follow standard conventions and report exact match (EM) scores using the reference answers included in each dataset. Table 2 shows our main results. We divide the table into three main sections: closed-book QA models, open-book QA models, and our implementation. The first two sections contain results from other papers, which we include for comparisons. The last section includes results from our proposed model, as well as our reimplementation of relevant baselines to control for our experimental setup.

Our reimplementation of T5-base provides strong baselines when the number of retrieved documents is set to 0 (no retrieval) and 1. From Table 2, we see that the setting of top-1 vastly improves performance over the setting with no retrieved documents, signifying the importance of retrieval for OpenQA tasks. When further increasing the top-$k$ documents to 50, the performance of the FiD models substantially improves over the top-1 retrieval, verifying the observation from (Izacard and Grave, 2021b) about the *importance of modeling the retrieved documents as a set*.

Comparing EMDR$^2$ with our reimplementation of FiD illustrates the benefit of our end-to-end training approach. The underlying model is similar in both cases, but the training method is different. FiD adopts a two-stage approach to first train the retriever and then the reader. We have three variants of FiD: (i) the reader and retriever are initialized with MSS training, (ii) the retriever is initialized with DPR training, which is the setting used in the original paper (Izacard and Grave, 2021b), and (iii) the retriever is initialized with MSS + DPR training from (Sachan *et al.*, 2021), as it further improves DPR recall. EMDR$^2$ outperforms all the variants by large margins on all the datasets.

The current best approach for training multi-document reader and retriever is FiD-KD (Izacard and Grave, 2021a). FiD-KD is a complex training procedure that requires multiple training stages and performs knowledge distillation with inter-attention scores. We take the results from the original paper when comparing our model with FiD-KD. EMDR$^2$ outperforms the reported numbers of FiD-KD by more than 2.5 points on NQ and TriviaQA to obtain new state-of-the-art results on these benchmarks.

In addition to better performance, EMDR$^2$ also has three other advantages compared to FiD-KD: (i) EMDR$^2$ is more efficient since it only uses 50 evidence documents, whereas FiD-KD leverages 100 documents; (ii) FiD-KD is based on a distillation approach which requires multiple cycles of retriever and reader training, while EMDR$^2$ only requires one cycle of end-to-end training; and (iii) FiD-KD relies on supervised initialization of the retriever to achieve its best performance. EMDR$^2$ is more robust to the retriever initialization, as demonstrated by state-of-the-art results even with unsupervised initialization of the retriever.

For the WebQ dataset, the training set size is much smaller compared to the other datasets (Table 5). Previous approaches such as RAG rely on supervised transfer (i.e., they finetune a model pre-trained on NQ) to obtain good results. In contrast, EMDR$^2$ improves over the results from this RAG model by 3.5 points *without the supervised transfer step*. This result demonstrates the applicability of our approach to the low-resource setting where we only have a limited number of training examples.

We also perform qualitative analysis of the model outputs, which is included in Appendix E.

## 3.5 Ablations

**Number of retrieved documents.** We investigate the performance of EMDR$^2$ and FiD as we vary the number of retrieved documents $K$ in Figure 2. We observe that when the number of retrieved documents is increased, both EMDR$^2$ and FiD improve in performance. When $K$ is small, the gap between EMDR$^2$ and FiD is larger. This indicates the efficacy of EMDR$^2$ in a more constrained setting where we can only retrieve a small number of documents (e.g., due to memory limitations).

**Retriever initialization.** We explore the effect of different parameter initialization strategies when training with EMDR$^2$: (i) unsupervised MSS pre-training, (ii) supervised retriever training (DPR), and (iii) MSS pre-training followed by supervised retriever training (MSS + DPR; Sachan *et al.* (2021)). Table 3 shows our results. We can see that on NQ, MSS pre-training being unsupervised leads to a lower initial retriever recall than DPR. After EMDR$^2$ training, the recall improves by 20% (highlighted in yellow cells). Training with DPR initialization leads to the same final recall as obtained by MSS

---

[8]We note that GPT-3 is not trained on the full training examples that we use, so the results are not directly comparable.

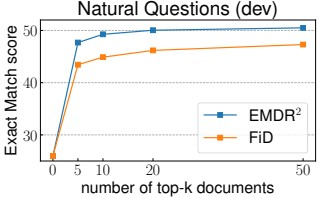 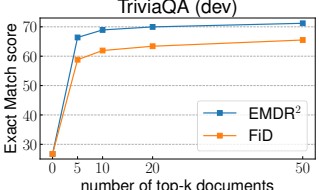 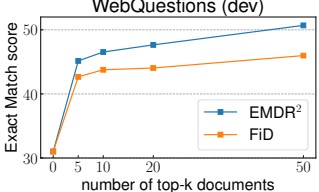

**Figure 2:** Performance on NQ, TriviaQA, and WebQ as we vary the number of retrieved documents.

| Retriever Initialization | Reader Initialization | NQ (dev) R@50 B.T. | NQ (dev) R@50 A.T. | NQ (dev) EM | TriviaQA (dev) R@50 B.T. | TriviaQA (dev) R@50 A.T. | TriviaQA (dev) EM | WebQ (dev) R@50 B.T. | WebQ (dev) R@50 A.T. | WebQ (dev) EM |
|---|---|---|---|---|---|---|---|---|---|---|
| MSS pre-training | MSS pre-training | 66.4 | 86.3 | 50.4 | 74.8 | 86.2 | 71.1 | 59.8 | 88.6 | 49.9 |
| MSS pre-training | T5 | 66.4 | 86.3 | 50.3 | 74.8 | 86.3 | 70.9 | 59.8 | 88.6 | 47.7 |
| DPR training | T5 | 82.3 | 86.3 | 50.0 | 83.2 | 86.2 | 70.5 | 84.2 | 88.6 | 49.0 |
| MSS + DPR | MSS pre-training | 84.5 | 86.3 | 50.5 | 85.3 | 86.3 | 71.2 | 85.0 | 88.6 | 49.9 |

**Table 3:** R@50 denotes the retrieval recall from the top-50 retrieved documents. B.T. and A.T. indicates R@50 score Before Training and After Training the model, respectively.

pre-training, *suggesting that DPR initialization of the retriever may not be an essential component to obtain good performance in OpenQA tasks.* Similar trends are also observed on TriviaQA and WebQ. Similarly, MSS + DPR initialization has a better initial recall but leads to a marginal or no improvements in answer extraction performance over MSS pre-training. Finally, we also observe that MSS pre-training also provides an improvement of 2 points in answer extraction on WebQ when compared to the T5 reader (shown in orange cells), highlighting its importance in the low-resource OpenQA tasks.

### 3.6 Alternative End-to-End Training Objectives

We compare EMDR$^2$ objective (Eq. 6) to two alternative formulations for end-to-end training.

In the first alternative formulation, when training the retriever parameters $\Phi$, we simply factorize $p(\mathcal{Z} \mid \boldsymbol{q}; \Phi) = \prod_{k=1}^{K} p(\boldsymbol{z}_k \mid \boldsymbol{q}; \Phi)$ to arrive at the following objective:

$$\mathcal{L}_{\text{alt-1}} = \log p(\boldsymbol{a} \mid \boldsymbol{q}, \mathcal{Z}; \Theta) + \sum_{k=1}^{K} \log p(\boldsymbol{z}_k \mid \boldsymbol{q}, \mathcal{Z}; \Phi).$$

| Method | top-$k$ | NQ | TriviaQA | WebQ |
|---|---|---|---|---|
| FiD | 50 | 47.3 | 65.5 | 46.0 |
| EMDR$^2$ | 50 | **50.4** | **71.1** | **49.9** |
| $\mathcal{L}_{\text{alt-1}}$ | 50 | 14.1 | 11.9 | 28.0 |
| $\mathcal{L}_{\text{alt-2}}$ | 50 | 49.9 | 69.6 | 28.8 |

**Table 4:** EM scores on the development set for alternative training objectives.

The second term in this objective is maximised by a uniform retrieval, in other words, by *removing* any discrimination between documents in the retriever. We include it to show the impact of an adversarial objective.

In the second formulation, for each retrieved document, we approximate its posterior under the assumption that we have a uniform prior over the set of retrieved documents: $\tilde{p}(\boldsymbol{z}_k \mid \boldsymbol{q}, \boldsymbol{a}, \mathcal{Z}_{\text{top-}K}; \Theta) \propto p(\boldsymbol{a} \mid \boldsymbol{q}, \boldsymbol{z}_k; \Theta) \times \frac{1}{K}$. We use this to train reader and retriever parameters as follows:

$$\mathcal{L}_{\text{alt-2}} = \log p(\boldsymbol{a} \mid \boldsymbol{q}, \mathcal{Z}; \Theta) + \mathbb{KL}(\mathbb{SG}\,(\tilde{p}(\boldsymbol{z}_k \mid \boldsymbol{q}, \boldsymbol{a}, \mathcal{Z}_{\text{top-}K}; \Theta)) \mid\mid p(\boldsymbol{z}_k \mid \boldsymbol{q}, \mathcal{Z}; \Phi)).$$

Intuitively, we try to match the probability of retrieving a document $\boldsymbol{z}_k$ with the "contribution" of that document to the generated answer $\boldsymbol{a}$, regardless of whether the retriever is relatively more or less likely to retreieve the document *a priori*.

Table 4 shows our results on the development set of NQ. We observe that training with the adversarial $\mathcal{L}_{\text{alt-1}}$ objective diverges, leading to poor performance, as expected. This shows that harming the retriever during training can significantly harm performance of the QA system. In contrast, although it disregards the estimated prior, the $\mathcal{L}_{\text{alt-2}}$ objective still improves over the FiD baseline for NQ and

TriviaQA. However, it still lags behind $\text{EMDR}^2$. On WebQ, the $\mathcal{L}_{\text{alt-2}}$ objective diverges and leads to a poor performance. We leave further analysis on the convergence of $\mathcal{L}_{\text{alt-2}}$ objective as a part of future work.

## 4   Related Work

Our work is based on end-to-end training of neural readers and retrievers, which we discuss in §1, §2, and §3. Here we instead focus on discussing previous work related to standalone neural retrievers, neural readers, and their application in other natural language processing tasks.

**Neural retrievers.** There are two broad classes of neural retrievers based on the number of embeddings computed for a document: dual encoders (Yih *et al.*, 2011, Lee *et al.*, 2019) and multivector encoders (Khattab and Zaharia, 2020, Luan *et al.*, 2021). Dual encoders store one embedding for each evidence document. Multivector encoders require multiple embeddings, which can be computationally expensive for large-scale retrieval. Due to the large size of the evidence document collection in OpenQA, our work uses the more efficient dual-encoder. Sachan *et al.* (2021) show that the performance of supervised dual encoders in OpenQA can be improved when pre-training with the Inverse Cloze Task for the high-resource setting or masked salient spans for the low-resource setting.

**Neural readers.** Neural readers output an answer given retrieved documents as its input. There are also two broad classes of neural readers: extractive and generative. Extractive readers (Clark and Gardner, 2018, de Masson d'Autume *et al.*, 2019, Wang *et al.*, 2019, Guu *et al.*, 2020, Karpukhin *et al.*, 2020) extract a span from a retrieved document to produce an answer. Generative readers (Izacard and Grave, 2021b), on the other hand, generates an answer conditioned on the retrieved documents.

**Other application areas.** In addition to question answering, retrieval-augmented methods have been successfully applied to other natural language processing tasks. In left-to-right language modeling, retrieving similar words from an external memory has been shown to improve perplexity (Khandelwal *et al.*, 2020, Yogatama *et al.*, 2021). In machine translation, retrieving domain-specific target language tokens has improved performance in domain adaptation (Khandelwal *et al.*, 2021). Finally, in dialog modeling, retrieving knowledge-informed text has helped improve factual correctness in the generated conversations (Fan *et al.*, 2021).

We provide a detailed comparison of $\text{EMDR}^2$ with some of the previous work in Appendix C and D.

## 5   Discussion

**Summary of contributions.** We presented $\text{EMDR}^2$, an end-to-end training method for retrieval-augmented question answering systems. We showed how to arrive at our training objective using the expectation-maximization algorithm. We demonstrated that $\text{EMDR}^2$ achieves state-of-the-art performance on three benchmark OpenQA datasets.

**Technical limitations.** $\text{EMDR}^2$ shares a few limitations with other retrieval-augmented question answering models. In particular, as evidence documents are stored in an uncompressed format, maintaining them and searching for relevant documents can be expensive (both in terms of compute and memory consumption). In our experiments, we only focused on open-domain question answering. It would be interesting to see how $\text{EMDR}^2$ performs for other text generation models as well. We also note that training is relatively resource-heavy (requiring 16 GPUs), potentially having environmental concerns.

**Potential negative societal impacts.** While $\text{EMDR}^2$ has the potential to improve language models in the low-resource setting (as demonstrated by our results on WebQ in §3.4), it could exhibit typical biases that are associated with large language models. For example, our model does not have an explicit mechanism to generate answers that are calibrated for fairness across all spectra. As a retrieval-augmented method, it also could be more prone to generating fake answers if an attacker manages to have access and modify information in the collection of evidence documents.

## Acknowledgements

The authors would like to thank the DeepMind Language team, Mila's students, and anonymous reviewers for providing us valuable feedback and useful suggestions about this work that helped us improve the paper.

## Funding Statement

DSS was supported by the Canada CIFAR AI Chair held by Prof. William Hamilton.

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
