| Dataset | Train | Filtered Train | Dev | Test |
|---|---|---|---|---|
| WebQuestions (WebQ) | 3,417 | 2,474 | 361 | 2,032 |
| Natural Questions (NQ) | 79,168 | 58,880 | 8,757 | 3,610 |
| TriviaQA | 78,785 | 60,413 | 8,837 | 11,313 |

**Table 5:** OpenQA dataset statistics. The training set is used for end-to-end training of the QA models whereas the filtered training set is used for supervised training of the retriever (i.e., for DPR experiments). The filtered set ignores those question-answer pairs where the evidence (Wikipedia) document retrieved using BM25 (Robertson and Zaragoza, 2009) does not align with the provided gold context documents. We leverage the filtered training set as provided by (Karpukhin et al., 2020).

## A Dataset Details

**Dataset statistics.** For validation, we randomly select approximately 10% examples from the training set. For all the datasets, we use the dataset splits from (Lee et al., 2019). We provide the size of the training, development, and test sets in Table 5.

**Pre-processing.** For TriviaQA experiments, following (Izacard and Grave, 2021a), we select human-annotated answers for training the QA model. We also filter out those questions whose answer length is more than 5 words. Overall, this filters out 2,362 examples from the training set.

**Dataset license and URLs.** All the datasets are open-source and widely used by the community. Below, we provide the URLs of the actual dataset source and their preprocessed version which is used in this work.

- NQ: *dataset*: `https://ai.google.com/research/NaturalQuestions/download`, *license*: `https://github.com/google-research-datasets/natural-questions/blob/master/LICENSE`
- TriviaQA: *dataset*: `http://nlp.cs.washington.edu/triviaqa/`, *license*: `https://github.com/mandarjoshi90/triviaqa/blob/master/LICENSE`
- WebQ: *dataset*: `https://github.com/google-research/language/tree/master/language/orqa#getting-the-data`, *license*: `https://nlp.stanford.edu/software/sempre/`
- Preprocessed version: We make use of NQ, TriviaQA, and evidence datasets as open-sourced by Karpukhin et al. (2020) here: `https://github.com/facebookresearch/DPR/blob/master/data/download_data.py`.

## B Additional Training Details

In addition to the details provided in §3.2, here, we provide further training details for reproducibility.

**BERT and Inverse Cloze Task (ICT).** We derive the implementations of BERT (Devlin et al., 2019) and ICT (Lee et al., 2019) from the open-source Megatron-LM toolkit.[9] For ICT, the dual-encoder retriever is initialized with BERT weights and then we train the model according to Lee et al. (2019). For training, we use Wikipedia paragraphs where we truncate the maximum length of a paragraph to 256 tokens. We list the settings and hyperparameters used for training BERT and ICT in Table 6.

**T5.** We derive the implementation of T5 (Raffel et al., 2020) language model from the open-source Megatron-LM toolkit (Shoeybi et al., 2019). We list the hyperparameters used for training T5 in Table 6. For consistency, we train T5 for the same number of steps and batch size as was done in the original paper. Additionally, we use BERT lowercase tokenization for both T5 and BERT.

---

[9] `https://github.com/NVIDIA/Megatron-LM`

| Hyperparameter | BERT | ICT | T5 | MSS |
|---|---|---|---|---|
| Dataset | Wikipedia, BookCorpus | Wikipedia | C4, Wikipedia, OpenWebText | Wikipedia |
| Num. Parameters | 110M | 220M | 220M | 440M |
| Hidden Size | 768 | 768 | 768 | 768 |
| Attention heads | 12 | 12 | 12 | 12 |
| Dropout | 0.1 | 0.1 | 0.1 | 0.1 |
| Optimizer | Adam | Adam | Adam | Adam |
| Batch Size | 256 | 4096 | 2048 | 64 |
| Training Steps | 1M | 100K | 1M | 82K |
| Warmup Ratio | 0.01 | 0.01 | 0.01 | 0.05 |
| Max. Learning Rate | 1e-4 | 1e-4 | 1e-4 | 2e-5 |
| Weight Decay | 1e-2 | 1e-2 | 1e-2 | 1e-1 |
| Learning Rate Decay | Linear | Linear | Linear | Linear |
| Gradient Clipping | 1.0 | 1.0 | 1.0 | 1.0 |

**Table 6:** Hyperparameters for training BERT, ICT, T5, and MSS models.

| Hyperparameter | NQ | TriviaQA | WebQ |
|---|---|---|---|
| Num. Parameters | 440M | 440M | 440M |
| Hidden Size | 768 | 768 | 768 |
| Attention heads | 12 | 12 | 12 |
| Dropout | 0.1 | 0.1 | 0.1 |
| Optimizer | Adam | Adam | Adam |
| Batch Size | 64 | 64 | 16 |
| Epochs | 10 | 10 | 20 |
| Warmup Ratio | 0.01 | 0.01 | 0.01 |
| Max. Learning Rate | 2e-5 | 2e-5 | 2e-5 |
| Weight Decay | 1e-1 | 1e-1 | 1e-1 |
| Learning Rate Decay | Linear | Linear | Linear |
| Gradient Clipping | 1.0 | 1.0 | 1.0 |
| Temperate ($\tau$) | 27.7 | 27.7 | 27.7 |

**Table 7:** Hyperparameters for finetuning on NQ, TriviaQA, and WebQ datasets.

**Unsupervised pre-training with masked salient spans (MSS).** For MSS training, we initialize the retriever of our model from the ICT weights and the reader from the T5 weights. We make use of the Stanza toolkit (Qi *et al.*, 2020) to segment evidence documents into sentences. We then extract named entities from these sentences using the NER model trained on the OntoNotes-5.0 dataset as provided by Stanza. These names entities are replaced by mask tokens. As the masked tokens correspond to special named entities, they are referred to as salient spans. The masked sentence is considered as the question to retrieve evidence documents and the reader is trained to generate the named entities corresponding to the masked salient spans with the help of retrieved documents. During retrieval, we ignore the evidence document from which the masked sentence was derived. We list the hyperparameters of MSS training in Table 6.

**Supervised training using the question-answer pairs.** We provide the training details in §3.2. We list the hyperparameters in Table 7. Apart from the number of epochs and batch size in WebQ, we use the same hyperparameters for all the experiments. For the temperature parameter ($\tau$) in Eq. 5, we follow Sachan *et al.* (2021) and set it as the square root of the hidden size.

**Training Time.** We run all of our experiments on a machine with 96 CPUs, 1.3TB physical memory, and 16 A100 GPUs. We use PyTorch (Paszke *et al.*, 2019) to implement our proposed model. With this hardware setup, our experiments on NQ and TriviaQA took approximately 25 hours to complete, while experiments on WebQ took roughly 8 hours to complete. Before supervised training, we also perform a one-time unsupervised MSS pre-training for 82,000 steps that took roughly 1 week.

| Method | R@5 after ICT | R@5 after MSS |
|---|---|---|
| REALM (Guu *et al.*, 2020) | 13.9 | 38.5 |
| EMDR$^2$ | 28.0 | 38.6 |

**Table 8:** Retrieval recall on the NQ development set after ICT and MSS pre-training.

| Method | Evidence Size | Evidence Dimension | GPU Memory (in FP16) |
|---|---|---|---|
| REALM (Guu *et al.*, 2020) | 13M | 128 | 3 GB |
| EMDR$^2$ | 21M | 768 | 30 GB |

**Table 9:** Comparison of evidence embeddings storage for retrieval.

## C   Unsupervised Pre-training and Comparisons with REALM

We make use of a couple of training techniques introduced in the REALM paper (Guu *et al.*, 2020): masked salient spans (MSS) pre-training and asynchronous evidence embedding update. There are similarities and differences in the way in which we apply these ideas to EMDR$^2$ training.

### C.1   ICT and MSS Pre-training

Both ICT and MSS are unsupervised techniques used to bootstrap the retriever so that it has a good initial recall.

We first initialize the retriever with ICT pre-training. For ICT, similar to REALM, we follow the settings in the ORQA paper (Lee *et al.*, 2019). We observe our Recall@5 to be much higher than that reported in the REALM paper (see Table 8). We believe that our choice of 768 dimensional embedding of each evidence document leads to better results when compared to the 128 dimensional embedding used in REALM.

We further pre-train with MSS once the retriever weights are initialized with ICT. We use a batch size of 64 and train for 82K steps using the EMDR$^2$ objective. In comparison, REALM uses a batch size of 512 and trains the model for 200K steps. Even with a much smaller batch size and training steps, EMDR$^2$ achieves similar Recall@5 after MSS training (Table 8). We hypothesize that with a large batch size and longer training, EMDR$^2$ would be able to further improve its recall. Another implementation detail is that EMDR$^2$ does not require the additional null document which was used in REALM.

For low-resource datasets such as WebQ, MSS pre-training also improves the performance of the FiD reader. As Table 3 illustrates, on WebQ, MSS pre-trained reader obtains a gain of more than 1 EM point over the T5 reader (shaded in orange color).

### C.2   Asynchronous Evidence Embedding Updates

The asynchronous evidence embedding updates are performed after every 500 steps of training and is similar to REALM with a couple of differences. In our work, asynchronous embedding updates is done both during MSS pre-training and supervised training, while in REALM it is performed only during MSS pre-training. The second difference, although a minor one, we needed to compute the embeddings of 21M evidence documents while REALM had to do this for 13M documents. We do this by having two process groups during training, one group trains the model on 8 GPUs while the other group performs evidence embedding computation on 8 GPUs in an asynchronous manner.

### C.3   Pre-computed Evidence Embeddings Storage for Retrieval

In Table 9, we provide some comparisons between REALM and EMDR$^2$ to showcase that the retrieval task is more challenging in our setting. Firstly, the size of evidence in REALM is 13M because each Wikipedia article is split into 288 wordpieces while the size of evidence in EMDR$^2$ is 21M as each Wikipedia article is split into 100 linguistic words. Second, the embedding dimension of each

evidence document in REALM is 128 while the embedding dimension of each evidence document in EMDR$^2$ is 768. Due to these factors, the memory required by REALM to store evidence embeddings (in FP16) is approximately 3 GB, while the memory required by EMDR$^2$ to store evidence embeddings (in FP16) is 30 GB. As the GPU RAM is constrained by its capacity (40 GB maximum in A100 GPUs), it was not possible to store the entire 30 GB embeddings in each GPU. Therefore, for online retrieval, we store the evidence embeddings in a distributed fashion over 16 GPUs and perform distributed asynchronous MIPS for fast retrieval.

## D   Comparison with Previous Work

Here we provide a discussion of how EMDR$^2$ is different from some of the previous work.

### D.1   Comparison with Hard EM and Reinforced Reader-Ranker Models

There are some similarities between EMDR$^2$ and $\mathcal{L}_{\text{alt-2}}$ to Hard EM (Min *et al.*, 2019) and Reinforced Reader-Ranker (R$^3$; Wang *et al.* (2018)), at the conceptual level even though they are not equivalent. Training with REINFORCE involves sampling from a policy network (i.e., the retriever in our case). We take a deterministic approach and take the top-K documents in both EMDR$^2$ and $\mathcal{L}_{\text{alt-2}}$. Compared to Hard EM, $\mathcal{L}_{\text{alt-2}}$ directly minimizes the KL divergence of the probability of a retrieved document with the probability of an answer given that document.

At the implementation level, there are many other differences between $\mathcal{L}_{\text{alt-2}}$ (and EMDR$^2$) with models in (Min *et al.*, 2019) and (Wang *et al.*, 2018). First, we would like to note that both these methods use TF-IDF and BM25 as their retrieval approach which are not trainable. In contrast, our work uses a dense retriever which is trained in an end-to-end manner. We list other differences in more detail below.

**Differences with Hard EM.**   Min *et al.* (2019) propose a hard EM approach to train an extractive reader model for QA tasks. The context document is assumed to contain multiple mentions of the correct answer. They propose an objective to train the reader. Specifically, during the training step, the model is trained using maximum marginal likelihood for the first $\tau$ steps and subsequently with their proposed logmax objective. In their open-domain QA experiments on TriviaQA and NQ, the retriever part is based on TF-IDF and BM25 and is non-trainable. Overall, their model is applicable to extractive readers without retriever training. In comparison, in EMDR$^2$, we train both the reader and retriever. As such, the hard EM approach is not directly applicable to our case.

**Differences with R$^3$.**   This paper involves three pipelined components: retriever, ranker, and reader. The retriever is BM25 based and is non-trainable. They jointly train the ranker and the reader. The ranker takes 100 documents from the retriever and selects one document to give as input to the reader (contrast this with our work that selects a set of documents). As this selection operation is non-differentiable, their model leverages policy gradient to train the ranker. They also propose a custom reward function based on the overlap of text between the extracted answer and the correct answer. The reader takes a single document as input. In contrast, our approach does not involve a ranker component, both the FiD reader and retriever are trainable, and our proposed objective function EMDR$^2$ is end-to-end differentiable.

### D.2   Comparison with Individual Top-K and Joint Top-K Models

**Comparison with Individual Top-K (Sachan *et al.*, 2021).**   Individual Top-K is another approach for end-to-end training but the difference is that it applies a single-document reader while EMDR$^2$ consists of a multi-document reader. Similar to previous methods like REALM and RAG, Individual Top-K objective function is also defined over multiple retrieved documents but is better optimized than them. As the performance of EMDR$^2$ is much better than Individual Top-K, EMDR$^2$ is a better modeling approach.

**Comparison with Joint Top-K (Sachan *et al.*, 2021).**   While both EMDR$^2$ and Joint Top-K are end-to-end training approaches for open-domain QA based on the FiD model, they are different in many ways. (i) *Different Objective Functions*: These approaches optimize different training objectives. To

achieve retriever training, Joint Top-K adds the retrieval probability score of the top-K documents to the unnormalized inter-attention scores. In this way, the reader pays more importance to those top-K documents with a higher retriever score. There is no explicit feedback from the reader to the retriever. In contrast, the second term in the training objective of $\text{EMDR}^2$ explicitly encourages the retriever to improve its predictions based on the agreement with the reader's answer-generation likelihood of a particular top-K document. (ii) *Task Performance*: $\text{EMDR}^2$ objective leads to a much improved end-to-end training algorithm. This is reflected by the performance gains over the FiD baseline. On NQ and TriviaQA, while $\text{EMDR}^2$ leads to 4.3 and 6.4 EM points improvements respectively, Joint Top-K obtains a much lower gain of 1 point improvement on NQ and no improvements on TriviaQA. This demonstrates that EMDR2 training leads to substantially better retrieval, that in turn leads to higher gains in answer generation. These results also illustrate that $\text{EMDR}^2$ is a much better end-to-end or joint training algorithm than Joint Top-K for the multi-document reader retriever approaches.

# E  Qualitative Analysis

In Table 10, we present some representative examples of the retriever output with both MSS pre-training and when the MSS pre-trained model is finetuned on NQ. We observe that after MSS pre-training, the top-1 outputs are related to the question but are not relevant enough to answer them. However, when the MSS pre-trained model is finetuned on NQ with $\text{EMDR}^2$, the retrieval accuracy improves with the top-1 documents being much more relevant to answer the question. The retriever's confidence score of the top-1 document also improves.

**Comparing retriever initializations.**   We analyze the reader's training loss when the retriever is either initialized with unsupervised MSS training or with first MSS pre-training followed by supervised DPR training (MSS + DPR). As indicated in Table 3, MSS pre-training being unsupervised has a lower accuracy while MSS + DPR retriever has a higher accuracy. However, as is also evident from the plots in Figure 4, retriever initialization has a marginal effect on the answer generation performance. We see that for NQ, for the first 1200 steps, the higher accuracy MSS + DPR retriever leads to a smaller training loss compared with the MSS retriever, after which the difference between the two training losses diminishes as the end-to-end training improves the accuracy of the MSS retriever. Similar trends are also observed for TriviaQA and WebQ but to a lesser extent.

**Visualizing reader and retriever losses.**   In Figure 3, we show the trajectories of the reader and retriever training losses when the model is initialized with MSS pre-training.

| Questions from NQ test | Answer | MSS Pre-training | EMDR[2] finetuned on NQ |
|---|---|---|---|
| what type of reaction occurs to form a dipeptide | peptide bond | probability=0.39 …Bornyl diphosphate synthase In enzymology, bornyl diphosphate synthase (BPPS) () is an enzyme that catalyzes the chemical reaction Bornyl diphosphate synthase is involved in the biosynthesis of the cyclic monoterpenoid bornyl diphosphate. As seen from the reaction above, BPPS takes geranyl diphosphate as its only substrate and isomerizes into the product, (+)- bornyl diphosphate. This reaction comes from a general class of enzymes called terpene synthases that … | probability=0.78 …Subsequent to this coupling reaction, the amine protecting group P and the ester are converted to the free amine and carboxylic acid, respectively. For many amino acids, the ancillary functional groups are protected. The condensation of the amine and the carboxylic acid to form the peptide bond generally employs coupling agents to activate the carboxylic acid. The Bergmann azlactone peptide synthesis is a classic organic synthesis for the preparation of dipeptides. … |
| when was the japanese videogame company nintendo founded | 23 September 1889 | probability=0.37 …contributed to the development of the following games. Creatures (company) Ape, Inc. was founded in March 1989 and Shigesato Itoi became its chief executive officer. Nintendo president Hiroshi Yamauchi had wanted to support new talent in game design. Liking Itoiś work, he proposed the idea of the company to Itoi and invested in it. Apeś staff included Tsunekazu Ishihara, who later became the Pokémon Companyś CEO, and Ashura Benimaru Itoh, a renowned illustrator. They began work on "Mother", which released in July. Its music was composed by Hip Tanaka, who later became the second CEO of Creatures … | probability=0.61 …Nintendo Co., Ltd. is a Japanese multinational consumer electronics and video game company headquartered in Kyoto. Nintendo is one of the world's largest video game companies by market capitalisation, creating some of the best-known and top-selling video game franchises, such as "Mario", "The Legend of Zelda", and "Pokémon". Founded on 23 September 1889 by Fusajiro Yamauchi, it originally produced handmade hanafuda playing cards. By 1963, the company had tried several small niche businesses, such as cab services and love hotels. Abandoning previous ventures in favour of toys in the 1960s … |

**Table 10:** Examples of top-1 retrieved documents from the NQ test when the model is pre-trained with Masked Salient Spans (MSS) or finetuned on NQ data. If the answer exists in the document it is highlighted in blue color, and the probability of the document (Eq. 5) is indicated in orange color.

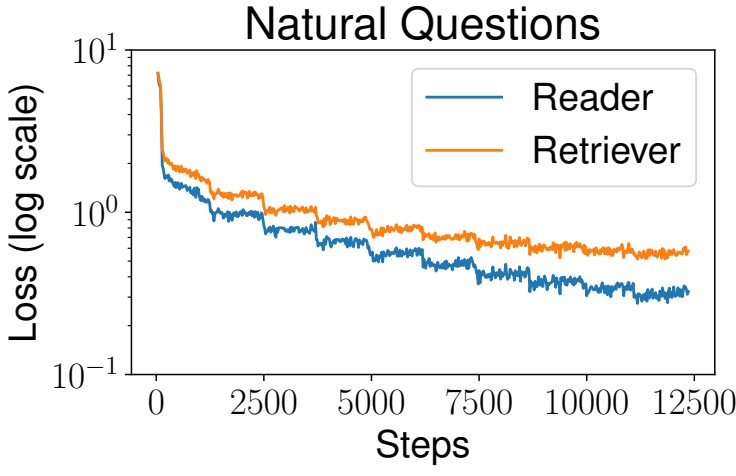

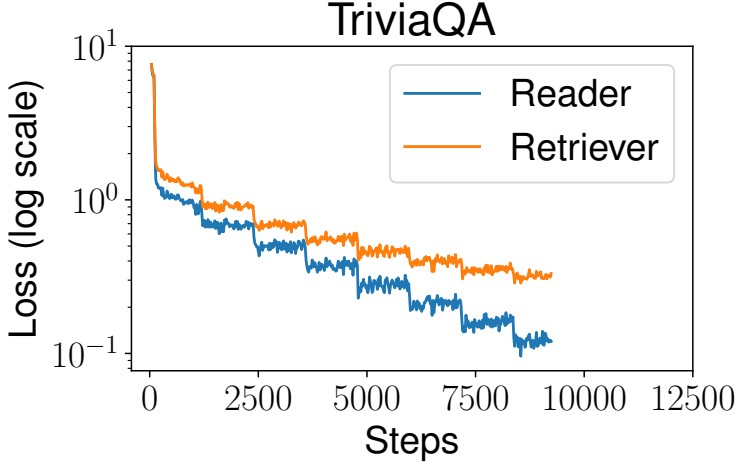

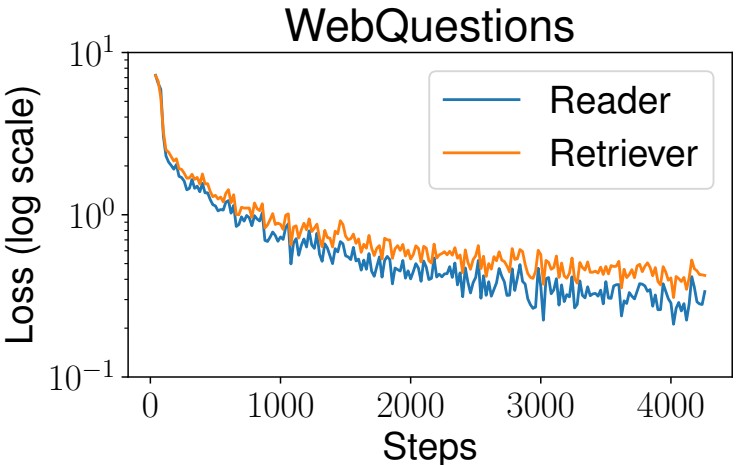

**Figure 3:** Reader and retriever training losses when the model is initialized with MSS pre-training.

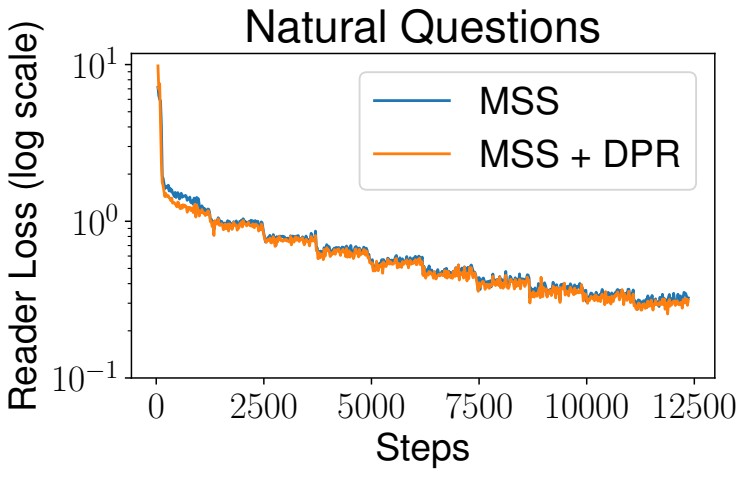

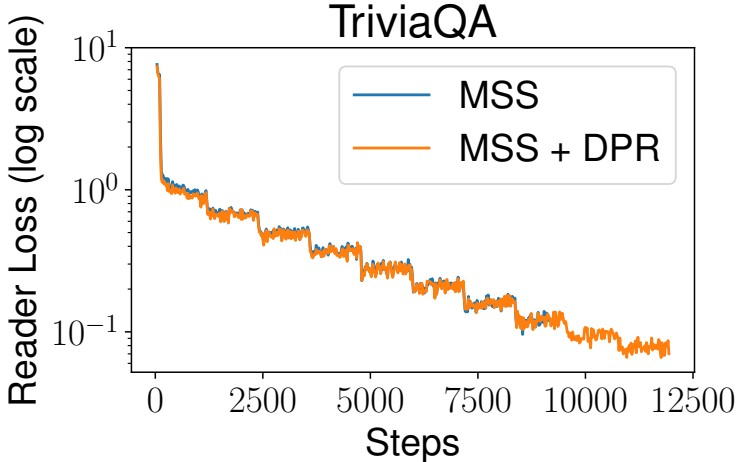

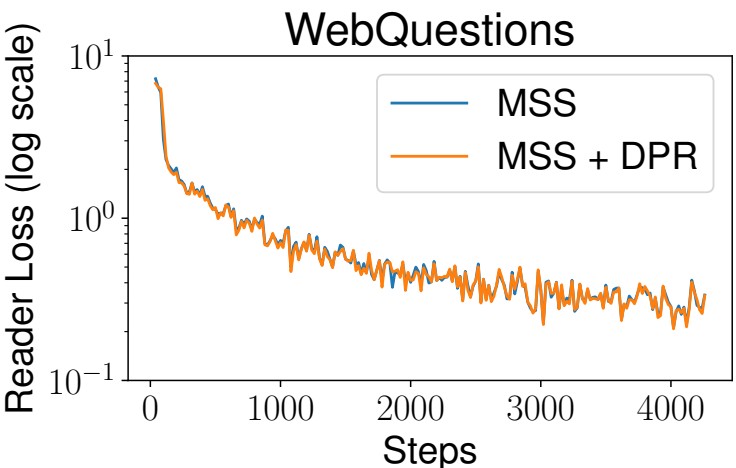

**Figure 4:** Reader training loss vs steps for NQ, TriviaQA, and WebQ when the retriever is either initialized by MSS pre-training or by MSS followed by supervised DPR training (MSS + DPR).