# OpenReview forum: "End-to-End Training of Multi-Document Reader and Retriever for Open-Domain Question Answering"
_NeurIPS.cc/2021/Conference — NeurIPS 2021 Poster_

### Official Review · Reviewer_aV2Y · 2021-07-07

**Rating:** 8
**Confidence:** 4

**Summary:**

The paper proposes end-to-end multi-document retriever-reader training for open-domain question answering. The proposed model is based on a pipeline approach where the retriever based on a dual encoder retrieves a set of documents from a large text corpus and a reader processes them and generates the answer. Different from previous approaches which independently train the retriever and the reader (Karpukhin et al, Izacard & Gave, and others), the paper jointly trains them using the EM algorithm. It is also different from previous work with joint training (Lee et al, Guu et al, and others) in that the generation of the answer is conditioned on multiple documents, not just a single one. In particular, this difference makes joint training more difficult, so the paper proposes to use the EM algorithm to iteratively update the parameters of the retriever and the reader. The paper also came up with clever initialization tricks and asynchronous refreshes of the MIPS index, which leads to successful training of the proposed model.
The proposed model is evaluated on widely-used open-domain QA benchmarks (NQ, TriviaQA, WebQ). The proposed model outperforms a set of other, recently-proposed models by a large margin, which is impressive given that the baselines are highly competitive.

**Limitations And Societal Impact:**

The paper adequately discusses limitations and societal impact. It would be also good to mention that training is relatively resource-heavy (requiring 16 GPUs), potentially having environmental concerns.

**Main Review:**

Strengths
- The problem is well-motivated, with clear distinctions from prior work.
- The model is well-executed, with the key method like EM algorithm, and tricks including initialization and asynchronous updates. All the descriptions of the models are very clear.
- The model achieves impressive results on competitive open-domain QA benchmarks.

Weaknesses
- I think the paper should discuss more thorough comparisons to REALM (Guu et al) because there are a lot of shared aspects. For example:
    * The asynchronous updates are almost identical.
    * The high-level idea of REALM pretraining is very similar to the initialization in this paper (using masked LM as sort of pseudo data to pretrain the model). Do you think these two methods have exactly the same role, or do you think there are features that your initialization can capture but REALM pretraining cannot?
  * The difference in answer generation being conditioned on single vs. multi documents (hence leading to the need for the EM algorithm) is clearer. Additional discussions I want to see are mainly about the above two points.
- Framing retrieval documents as a latent variable is first done by Lee et al. 2019. It should be more highlighted.


Questions
- Asking based on ablation in Table 3
    * What is P@50? What “retrieval accuracy” means is unclear. Do you mean recall accuracy? (The model gets a credit if at least one of top-50 docs contain the answer text.) In that case, it might be better to say “R@50” or “Rec@50” then, and clarify in the caption?
    * It looks like DPR-style supervised training does not impact the performance very much based on this metric. Have you compared these two methods based on the EM score?
- Have you tried a baseline where you initialize the retriever and the reader using MSS pretraining, but then train them separately, in a pipeline manner? I think this is an important baseline.


Minor comments
- L19: “train a language model”: I am not sure if it is necessary to train a “language model”?
- L32: I would cite Karpukhin et al for the pipeline training as it is the first paper trying it, and Izacard & Grave mainly took the model from Karpukhin et al.
- Table 1: It might be good to add a column “requiring document supervision” which makes the proposed model more appealing - I think this column will be checked only for REALM and the proposed model.
- Section 3.2: Which T5 variant was used for the reader? I am guessing it is T5-base, but it isn’t clearly mentioned in the paper.
- Section 3.3: I think the terms “parametric” and “semi-parametric” are confusing, and are not used in any prior work. What about “closed-book” vs. “open-book” (following terms in Roberts et al) or “no retrieval” vs. “retrieval”?


Conclusion & Overall recommendation

The paper proposes a model for open-domain QA, where the answer is generated conditioned on multiple documents jointly, and the retriever and the reader are trained end-to-end. The proposed idea is new and substantially different from prior work; the model is well-executed; the experiments are comprehensive, showing impressive results on competitive open-domain QA benchmarks. I have minor concerns about more comparisons with prior work and additional baselines, which are relatively easy to address. Overall, the quality of the paper is substantially higher than the bar to be published.

**Time Spent Reviewing:**

2

---

> ### Author Response · Authors · 2021-08-10
> **Authors Response**
>
> We sincerely thank the reviewer for providing the review and their thoughtful comments.
>
> ### Response to questions
>
> **P@50 and retrieval accuracy**
>
> Thank you for pointing this out. Yes, as you mentioned, we meant *recall accuracy* i.e, the model gets a credit if at least one of the top-50 documents contain the answer. We will make this correction in Table 3 in the paper.
>
> **Impact of DPR initialization**
>
> Yes, we observed that with $\text{EMDR}^2$ training, the MSS-based initialization of the retriever leads to the same final Recall@50 which is obtained after the DPR-based initialization. Considering the example of NQ dev, the MSS pre-training has a lower initial Recall@50 of 66.4, which is expected as it is unsupervised, while DPR training being specific to NQ has a higher Recall@50 of 82.3. However, after the EMDR^2 training, both MSS and DPR initializations lead to the same final Recall@50 of 86.3 . Due to this, both these methods lead to a very close answer extraction score of 50.4 EM. This suggests that DPR initialization of the retriever is not necessary when performing end-to-end training with $\text{EMDR}^2$.
>
> **MSS training as a baseline**
>
> *MSS retriever, MSS reader*: In this, we use the outputs of the MSS initialized retriever to train the MSS reader. More specifically, only the MSS initialized FiD reader is trained from the outputs of the MSS retriever in a pipelined manner. The results of this baseline are reported under the section “Our Implementation” in Table 2 under the row FiD (MSS retriever, MSS reader). As the Recall@50 of the MSS retriever is low, the answer extraction performance of the FiD reader suffers.
>
> *MSS+DPR retriever, MSS reader*: In this, the retriever is initialized with MSS training and is further trained with the DPR approach. The MSS+DPR retriever is more accurate than the MSS retriever (i.e, Recall@50 of the MSS+DPR retriever is 84.5 while that of the MSS retriever is 66.4 on NQ dev). When the MSS reader is trained with the outputs of MSS+DPR retriever in a pipelined fashion, the answer extraction performance on NQ is (Dev: 48.2 EM, Test: 49.3 EM). When further trained with $\text{EMDR}^2$, the answer extraction performance of the MSS+DPR retriever improves to (Dev: 50.4 EM, Test: 52.5), as can also be seen in Table 3. In the next version of the paper, we will include the result of the MSS+DPR retriever, MSS reader baseline trained in a pipelined manner for all the datasets in Table 2.
>
> ### Response to minor comments
> Thank you for providing these useful suggestions. We will incorporate them in the next version of the paper. For the experiments, we use the T5-base model. We will make it more clear in Section 3.2.
>
>
> ### Comparisons to REALM
> We will add a more detailed discussion about similarities and differences to REALM in the next version of the paper. We will also highlight that framing retrieval documents as latent variables is first done by (Lee et al., 2019) in the paper. We provide an outline of these changes below.
>
> We make use of a couple of training techniques introduced in the REALM paper: masked LM pre-training and asynchronous evidence embedding update. There are similarities and differences in the way in which we apply these ideas to $\text{EMDR}^2$ training:
>
> **ICT training and Masked LM pre-training**
>
> Both these unsupervised techniques are used to bootstrap the retriever so that it has a good initial recall.
>
> We first initialize the retriever with ICT training. For ICT, similar to REALM, we follow the settings in the ORQA paper (Lee et al., 2019). We observe our Recall@5 to be much higher than that reported in the REALM paper (see Table below). We believe that the choice of 768 dimensional embedding of each evidence document leads to better results when compared to the 128 dimensional embedding used in REALM.
>
> For model pre-training with masked salient spans (MSS), the retriever weights are initialized with the ICT training. In this work, we use a batch size of 64 and train for 82K steps using the $\text{EMDR}^2$ objective. However, the REALM paper uses a batch size of 512 and trains the model for 200K steps. Even with a much smaller batch size and number of training steps, $\text{EMDR}^2$ achieves similar Recall@5 after MSS training. We hypothesize that with a larger batch size and longer training, $\text{EMDR}^2$ would be able to further improve Recall@5. Another implementation detail is that $\text{EMDR}^2$ does not need the additional null document which was used by REALM.
>
> Dataset: NQ Dev | Recall@5 after ICT training | Recall@5 after MSS pre-training
> -----------------------|--------------------------------------|--------------------------------------------
> REALM     | 13.9 | 38.5
> $\text{EMDR}^2$ | 28.0 | 38.6
>
>
> For low-resource tasks such as that of WebQ, MSS pre-training also improves the performance of the FiD reader. As Table 3 illustrates, on WebQ, MSS pre-trained reader obtains a gain of more than 1 EM point over the T5 reader (shaded in orange color).
>
> **Asynchronous Evidence Embedding Updates**
>
> The asynchronous evidence embedding updates are performed after every 500 steps of training and is similar to REALM with a couple of differences. In our work, asynchronous embedding updates is done both during MSS pre-training and supervised training, while in REALM it is performed only during MSS pre-training. The second difference, although a minor one, is that in our paper, we need to compute the embeddings of 21M evidence documents while REALM had to do this for 13M documents. We do this by having two process groups during training, one group trains the model on 8 GPUs while the other group performs evidence embedding computation on 8 GPUs in an asynchronous manner.
>
> **Pre-computed Evidence Embeddings Storage for Retrieval**
>
> We provide some comparisons between $\text{EMDR}^2$ and REALM to showcase that the retrieval task is more challenging in the $\text{EMDR}^2$ setting compared to REALM.
>
> Number of evidence documents in REALM = 13M (because each wikipedia article is split into 288 wordpieces)
> Number of evidence documents in $\text{EMDR}^2$ = 21M (because each Wikipedia article is split into 100 linguistic words)
>
> Embedding dimension of each evidence document in REALM: 128
> Embedding dimension of each evidence document in $\text{EMDR}^2$: 768
>
> Memory required by REALM to store evidence embeddings (in FP16) ~ 3 GB
> Memory required by $\text{EMDR}^2$ to store evidence embeddings (in FP16) ~ 30 GB
>
> Due to GPU RAM constraints, it is not possible to store the entire 30 GB embeddings in each GPU. Therefore, for online retrieval, we store the evidence embeddings in a distributed fashion over 16 GPUs and perform distributed asynchronous MIPS for fast retrieval.

---

### Official Review · Reviewer_gGYZ · 2021-07-11

**Rating:** 6
**Confidence:** 3

**Summary:**

This paper focuses on the open-domain question answering task. They use the generate framework which first retrieve relevant documents for the given question and then generate the answer based on the question and the retrieved documents. They model the retrieved documents in the latent space. Previous studies usually consider only a single document as the latent variable. In this paper, they consider to use a set of documents as the latent variables. They then apply an expectation-maximization algorithm to update the reader and the retriever. The reader is optimized to generate the answer based on the question and the retrieved set of documents. The retriever is updated using posterior estimation of the retrieved results given the question and the answer. They conduct experiments on three open domain question answering tasks. The experimental results show that their approach outperforms other baselines by a large margin (2%-3% absolute improvement). They conducted ablation studies to show the performance of the model with different number of retrieved documents. They also compared different retriever initialization  strategies and showed that masked salient spans pretraining is helpful.

**Ethics Review Area:**

["I don’t know"]

**Limitations And Societal Impact:**

The model may have typical biases caused by the large language models. Another potential limit is the model could be attacked to generate fake answers when the evidence documents are modified.

**Main Review:**

The originality:
The authors propose to model a set of retrieved document instead of single document in the latent space. This is different from previous work such as REALM. Based on this intuition, the EM algorithm is straightforward to come up with and the objective is similar to the one in REALM.

Quality:
Their experimental results show that their model can significantly outperforms all the baselines. The also conducted a set of ablation study which show that their model performs consistently good with different number of retrieved documents. But they did not do the experiments to show that it is important to model a set of retrieved documents instead of a single one. For example, they could conduct preliminary experiments to show how many questions need to be answered based on multiple documents, and do the analysis to show that their model can performs much better especially on the cases that required multi-document modeling. Without showing this,  their motivation is not well supported by the experiments.

Clarity:
The paper is well-written and easy to understand. The approach is intuitive and straightforward. The authors will release the code for further research and reproducibility.

Significance:
The authors achieve state-of-the-art performance on three popular dataset for open domain question answering, which is significant.


**Time Spent Reviewing:**

3

---

> ### Author Response · Authors · 2021-08-10
> **Authors Response**
>
> We sincerely thank the reviewer for providing the review and their thoughtful comments.
>
> Regarding your comment about experiments on modeling a set of documents vs. a single document, previous works have established that models that only make use of a single document (e.g., REALM, DPR, RAG) underperform models that use a set of documents (e.g., FiD). Also, as noted by Reviewer 1, *"... multi-document readers like FiD have become the mainstream in many QA tasks, it is necessary to have a joint-training algorithm like REALM be extended to these readers".*
>
> We also have performed an apple-to-apple empirical evaluation to corroborate this observation. When using a single retrieved document (i.e., top-K = 1), the exact match scores on NQ are 37.3 (dev set) and 38.4 (test set). In the multiple documents setting (top-K = 50), the scores are 47.3 (dev set) and 48.3 (test set). We can see that modeling a set of retrieved documents is important and produces significantly better results. In addition, Figure 2 illustrates that with more retrieved documents, the performance keeps improving. We will add the results comparing top-K = 1 and top-K = 50 for all datasets in the next version of the paper.

---

### Official Review · Reviewer_6UDc · 2021-07-14

**Rating:** 6
**Confidence:** 4

**Summary:**

This paper proposes EMDR2, an end-to-end differentiable training method of multi-document reader and retriever for open-domain question answering. Based on expectation-maximization algorithm, EMDR2 iteratively estimates the set of relevant documents for a given question and uses the estimated set to update the parameters of retriever and reader. EMDR2 achieves new SOTA performance for models of comparable size on three benchmark OpenQA datasets.
Contributions:
-	An **EM-based** end-to-end training method of **multi-document** reader and retriever is proposed, which achieves SOTA performance on three benchmark OpenQA datasets.
-	The observation that EMDR2 implementations with different retriever initializations achieve similar final retrieval accuracies may be instructive.


**Limitations And Societal Impact:**

Yes

**Main Review:**

Originality:
-	This is new end-to-end training method for open-domain QA. Although the differences from previous work (such as REALM, RAG, and FiD) are clear, EMDR2 looks like a better combination of their techniques. The new contributions compared to Joint Top-k (Sachan et al., 2021) in Table 3 seem to be somewhat incremental and conventional.
-	Although a few end-to-end training methods for OpenQA are mentioned in introduction, there is no more comprehensive and in-depth literature review for such methods in related work.

Quality:
-	The handling of technology and its terminology sounds correct and makes sense, but some intuitive explanations are lacked, e.g., the assumption on line 129 and the reason for using stop-gradient operator.
-	The work is complete and its empirical results seem to be impressive. But the necessity of considering multiple documents during end-to-end training and inference for single-hop questions from the evaluated three datasets is not intuitive explained and experimentally proved. Maybe a variant of EMDR2 whose reader only takes as input a question and one document (i.e., Z_reader = Z_{top-1}) should be considered. The result comparison between the variant and EMDR2 could prove such necessity, and the comparison between it and the Individual Top-k (Sachan et al., 2021) can show the effectiveness of updating the retriever using the posterior.

Clarity:
-	The paper is easy to follow, and an expert reader may reproduce the proposed methods and results with some effort. However, the core motivation and main contributions are not highlighted, which brings ambiguity to the proposed method's novelty.
-	The conclusion on lines 47 to 48 is inconsistent with that on lines 260 to 261.
-	Should the metric P@50 be Recall@50? It is abnormal for P@50 having values greater than 80%, which means there are more than 40 relevant documents for a question from the three datasets.

Significance:
-	EMDR2 provides a better way for end-to-end training of multi-document retriever and reader than Joint Top-k (Sachan et al., 2021), and achieves SOTA performance on three benchmark OpenQA datasets. It may bring improvements in training methods for generative OpenQA and other retrieval-augmented NLG tasks.
-	The observation that EMDR2 implementations with different retriever initializations achieve similar final retrieval accuracies may be instructive.


**Time Spent Reviewing:**

12

---

> ### Author Response · Authors · 2021-08-10
> **Authors Response**
>
> We sincerely thank the reviewer for providing the review and their thoughtful comments.
>
> **Comparison between $\text{EMDR}^2$ and Joint Top-K (Sachan et al. 2021)**
>
> While both $\text{EMDR}^2$ and Joint Top-K are end-to-end training approaches for open-domain QA based on the FiD model, they are different in many ways.
>
> i) *Different Objective Functions*: These approaches optimize different training objectives. To achieve retriever training, Joint Top-K adds the retrieval probability score of the top-K documents to the unnormalized inter-attention scores (Eq. 8 in Sachan et al. 2021). In this way, the reader pays more importance to those top-K documents with a higher retriever score. There is no explicit feedback from the reader to the retriever. In contrast, the second term in the training objective of $\text{EMDR}^2$ explicitly encourages the retriever to improve its predictions based on the agreement with the reader’s answer-generation likelihood of a particular top-K document.
>
> ii) *Task Performance*: $\text{EMDR}^2$ objective leads to a much improved end-to-end training algorithm. This is reflected by the performance gains over the FiD baseline. On NQ and TriviaQA, while $\text{EMDR}^2$ leads to 4.3 and 6.4 EM points improvements respectively, Joint Top-K obtains a much lower gain of 1 point improvement on NQ and a drop of 0.2 point on TriviaQA. This demonstrates that $\text{EMDR}^2$ training leads to substantially better retrieval, that in turn leads to higher gains in answer generation. These results also illustrate that $\text{EMDR}^2$ is a much better end-to-end or joint training algorithm than Joint Top-K for the multi-document reader retriever approaches.
>
>
> **Intuitive explanations about assumptions and the stop-gradient operator**
>
> We will add them in the next version of the paper. The intuition behind our assumption that the probability of the set is maximized if the sum of the probability of each document in the set is maximized is that we assume each element of the set contributes independently, which greatly simplifies the computation to find the maximum of the set.
>
> The stop-gradient operator in the second term of $\text{EMDR}^2$ which causes it to not backpropagate through the single-document FiD reader has several benefits. First, the FiD reader is trained from the first term of the $\text{EMDR}^2$ objective in which its likelihood is conditioned on all the retrieved documents, similar to how the reader is used at test time. It also makes training faster as backward pass which is 3x more expensive than the forward pass is not needed, which in turn reduces usage of GPU RAM as intermediate activations need not be saved.
>
> **Results with single retrieved document (top-K = 1)**
>
> We have performed experiments to evaluate the performance of $\mathcal{Z}_{top-1}$ based on your suggestion. When using a single retrieved document (i.e., top-K = 1), the exact match scores on NQ are 37.3 (dev set) and 38.4 (test set). In the multiple documents setting (top-K = 50), the scores are 47.3 (dev set) and 48.3 (test set). We can see that modeling a set of retrieved documents is important and produces significantly better results. In addition, Figure 2 illustrates that with more retrieved documents, the performance keeps improving. We will add the results comparing top-K = 1 and top-K = 50 for all datasets in the next version of the paper.
>
> **Comparison with Individual Top-K (Sachan et al. 2021)**
>
> Individual Top-K is another approach for end-to-end training but the difference is that it applies a single-document reader while $\text{EMDR}^2$ consists of a multi-document reader. Similar to previous methods like REALM and RAG, Individual Top-K objective function is also defined over multiple retrieved documents and is better optimized than them. As the performance of $\text{EMDR}^2$ is much better than Individual Top-K, $\text{EMDR}^2$ is a better modeling approach.
>
>
> **Response to the clarity section of the review**
>
> *Core Motivation and Contributions*: We will better clarify our core motivation and contributions to help readers understand the novelty of our proposed approach. Our motivation is to provide an end-to-end training algorithm for multi-document reader retriever QA model, which forms the basis of many state-of-the-art open-domain QA models. To the best of our knowledge, our approach is the first method to do this, which also results in new new state-of-the-art results on three widely used datasets of Natural Questions, TriviaQA, and WebQuestions, obtaining absolute gains of 2-3% over the previously best published results on these tasks. In addition to extensive empirical validations of the efficacy of our proposed method, other contributions of the paper include findings that supervised (pre-) training of the retriever is not necessary to achieve good results when a model is trained with our algorithm as well as an open-source release of our code. We will improve the paper to highlight the novelty of our proposed method according to your suggestions.
>
>
> *The conclusion on lines 47 to 48 is inconsistent with that on lines 260 to 261*: Could you please elaborate why the conclusion on lines 47-48 is inconsistent with 260-261? In 47-48, we state that $\text{EMDR}^2$ achieves high accuracy with unsupervised pre-training initialization, suggesting that supervised training might not be necessary. In 260-261, we mention that training with DPR (supervised training) leads to the same final accuracy as MSS pre-training (unsupervised pre-training), which indicates that the supervised training component might not be necessary. Based on your further input, we will try to make this more clear in the next version of the paper.
>
>
> *Should the metric P@50 be Recall@50?*: Yes, it should be Recall@50. Thanks for pointing this out. We will fix this in the next version of the paper.
>
>
> **Discussion with previous joint and end-to-end training methods for the reader-retriever architecture**
>
> We will add detailed discussions in the next version of the paper.

---

### Official Review · Reviewer_p6SD · 2021-07-16

**Rating:** 6
**Confidence:** 4

**Summary:**

This paper makes end-to-end training work for multi-document readers (e.g., FiD style readers). The key idea is an EM algorithm to update the reader and retriever given the top-k selection of previous iteration models. The proposed approach shows good performance on multiple open-domain QA benchmarks.

**Main Review:**

This paper makes end-to-end training work for multi-document readers (e.g., FiD style readers). The key idea is an EM algorithm to update the reader and retriever given the top-k selection of previous iteration models.

Although I think that the novelty of the EM algorithm is a bit limited (mathematically, the Hard-EM, REINFORCE, REALM and iteratively updating are closely related), the proposed approach demonstrates a clear advantage on multiple open-domain QA benchmarks. Also given that multi-document readers like FiD have become the mainstream in many QA tasks, it is necessary to have a joint-training algorithm like REALM be extended to these readers. Therefore I still would like to vote for acceptance.

It will be great if the following two questions can be addressed in the author rebuttal:

(1) To deal with the posterior, in Eq. 4 the authors make an approximation, which reduces the multi-document reader to single-document reader. On the other hand, could I understand the L_{alt-2} objective in Section 3.6 is equivalent to either Hard-EM [1] or REINFORCE [2]. If the answer is NO, detailed discussions about the differences from [1,2] should be added; if YES, this L_{alt-2} should be added as a baseline to Table 2 under "Our Implementation". In whichever case, I think the discussion of L_{alt-2} should be highlighted in Section 3.4, because it is related to a straightforward way of applying what has been developed by the community [1,2], while can form an apple-to-apple comparison to the proposed approximation.

[1] Min et al., 2019. A discrete hard em approach for weakly supervised question answering.

[2] Wang et al., 2018. R3: Reinforced ranker-reader for open-domain question answering.

(2) The reader of EMDR^2 is based on T5. If I am correct, the RAG and FiD works use BART instead. I am wondering how much benefit is caused by switching BART to T5. Therefore it would be helpful if EMDR^2 can be applied to the BART reader for comparison.

Typo:

Line 103: Eq. 3 (should be the equation in Line 101?)



**Time Spent Reviewing:**

2

---

> ### Author Response · Authors · 2021-08-10
> **Authors Response**
>
> We sincerely thank the reviewer for providing the review and their thoughtful comments.
>
> ### Comparison of $\mathcal{L}_{alt2}$, Hard EM, and REINFORCE
> Regarding your first question about the connections of  $\mathcal{L_{alt2}}$  to Hard EM and REINFORCE, there are some similarities at the conceptual level even though they are not equivalent. Training with REINFORCE involves sampling from a policy network (i.e., the retriever in our case). We take a deterministic approach and take the top-K documents in $\mathcal{L_{alt2}}$. Compared to Hard EM, $\mathcal{L_{alt2}}$ directly minimizes the KL divergence of the probability of a retrieved document with the probability of an answer given that document. We will add the results of $\mathcal{L_{alt2}}$ for TriviaQA and WebQuestions and include them in Table 3 in the next version of the paper according to your suggestion.
>
> At the implementation level, there are many other differences between $\mathcal{L_{alt2}}$ (and $\text{EMDR}^2$) with models in [1] and [2]. First, we would like to note that both [1] and [2] use TF-IDF and BM25 as their retrieval approach which are not trainable. In contrast to these works, our work uses a dense retriever which is trained in an end-to-end manner. We list other differences in more detail below.
>
> **Differences with “Min et al., 2019”**
>
> Min et al., 2019, propose a hard EM approach to train an extractive reader model for QA tasks. The context document is assumed to contain multiple mentions of the correct answer. They propose an objective to train the reader. Specifically, during the training step, the model is trained using maximum marginal likelihood for the first $\tau$ steps and subsequently with their proposed (logmax) objective. In their open-domain QA experiments on TriviaQA and Natural Questions, the retriever part is based on TF-IDF and BM25 and is non-trainable. Overall, their model is applicable only to extractive readers with no retriever training. In comparison, in $\text{EMDR}^2$, we train both the reader and retriever. As such, their hard EM approach is not directly applicable to our case.
>
> **Differences with “Wang et al., 2018”**
>
> This paper involves three pipelined components: retriever, ranker, and reader. The retriever is BM25 based and is non-trainable. They jointly train the ranker and the reader. The ranker takes 100 documents from the retriever and selects one document to give as input to the reader (contrast this with our work that selects a set of documents). As this selection operation is non-differentiable, their model leverages policy gradient to train the ranker. They also propose a custom reward function based on the overlap of text between the extracted answer and the correct answer. The reader takes a single document as input. In contrast, our approach does not involve a ranker component, both the FiD reader and retriever are trainable, and our proposed objective function $\text{EMDR}^2$ is end-to-end differentiable.
>
> **Relation between $\text{EMDR}^2$ objective (Eq. 5) and $\mathcal{L}_{alt2}$**
>
> Compared to the previous works, $L_{alt2}$ is a novel training algorithm proposed in our paper and is also closely related to Eq. 5, the $\text{EMDR}^2$ training objective. The difference between these two lies in the second term, i.e, the manner in which the retriever parameters get updated. While we present a derivation of Eq. 5 using the posterior involving the answer, $\mathcal{L}_{alt2}$ first normalizes the $p(a \mid q, z_i)$ for all the $z_i$ to obtain a (valid) probability distribution ($\tilde{p}$) and then minimizes the KL-divergence such that the retriever probability distribution gets matched to the ($\tilde{p}$) distribution.
>
> ### BART vs T5 readers
> For your second question, we first would like to clarify that the FiD work (Izacard and Grave, 2021) uses T5 weights. In our paper, we take a similar approach and use the T5-base configuration (220M parameters). In Table 2, we compare with the corresponding results reported in the FiD paper.
>
> As you noted, the RAG model uses BART-large which has 400M parameters. Due to GPU memory limitations, we are currently not able to do $\text{EMDR}^2$ experiments with large model configurations. However, our best result using the T5 base model already outperforms the best reported result from RAG.
>
> As a side note, we also would like to point out that (Sachan et al., 2021) report results using the T5 large configuration optimizing the same objective as the RAG model. They report a gain of 3.5 EM points on NQ in answer extraction performance when compared to the RAG model, illustrating that initialization with T5 weights is better than BART on open-domain QA.
>
> **References**:
>
> Izacard, G. and Grave, E. (2021). Leveraging passage retrieval with generative models for open domain question answering. In Proceedings of the 16th Conference of the European Chapter of the Association for Computational Linguistics: Main Volume.
>
> Sachan, D. S., Patwary, M., Shoeybi, M., Kant, N., Ping, W., Hamilton, W. L., and Catanzaro, B. (2021). End-to-end training of neural retrievers for open-domain question answering. In Joint Conference of the 59th Annual Meeting of the Association for Computational Linguistics and the 11th International Joint Conference on Natural Language Processing (ACL-IJCNLP).

---

### Decision · Program_Chairs · 2021-09-27

**Decision:**

Accept (Poster)

**Comment:**

This paper proposes a new algorithm for training of multi-document question answering model containing a reader and a retriever. The key idea is to introduce latent variables over the sets of retrieved documents, employ the EM algorithm to approximately estimate the variables, and update the parameters of the reader and the retriever. Experimental results show that the proposed approach outperforms the baselines for about 2-3% on three benchmark datasets.

Pros
* The problem studied is important.
* The paper is generally clearly written.
* The proposed approach appears to be reasonable and sound.
* Experimental results show the efficacy of the proposed approach. The improvements are significant.

Cons
* There were related models proposed in the past. The idea of the work is not very novel. The paper mainly demonstrates that a new way of combining the existing ideas does work better.
* There are some details that are not explained very clearly. The authors have promised to make them clearer in the revised version.
* The authors have also successfully addressed many of the issues pointed out by the reviewers in the rebuttal.

Overall this is a solid work. We think that the contribution to ML is mainly from the empirical results.